# REACT study protocol: resilience after the COVID-19 threat (REACT) in adolescents

Alicia Joanne Smith ![ORCID] ,[1,2] Laura Moreno-López ![ORCID] ,[1] Eugenia Davidson,[1,3] Maria Dauvermann ![ORCID] ,[1,4] Sofia Orellana,[1] Emma Soneson,[1] Konstantinos Ioannidis,[1,5] Muzaffer Kaser,[1,5] Anne-Laura van Harmelen[1,6]

[1]Department of Psychiatry, University of Cambridge, Cambridge, UK
[2]MRC Cognition and Brain Sciences Unit, University of Cambridge, Cambridge, UK
[3]College of Medicine and Health, University of Exeter, Exeter, UK
[4]McGovern Institute for Brain Research, Massachusetts Institute of Technology, Cambridge, Massachusetts, USA
[5]Cambridgeshire and Peterborough NHS Foundation Trust, Cambridge, UK
[6]Education and Child Studies, Leiden University, Leiden, Netherlands

**Correspondence to**
Ms Alicia Joanne Smith;
alicia.smith@mrc-cbu.cam.ac.uk

## ABSTRACT

**Introduction** COVID-19-related social isolation and stress may have significant mental health effects, including post-traumatic stress, anxiety and depression. These factors are thought to disproportionately affect populations at risk of psychopathology, such as adolescents with a history of childhood adversity (CA). Therefore, examining which factors may buffer the impact of COVID-19-related stress and isolation in vulnerable adolescents is critical. The Resilience After the COVID-19 Threat (REACT) study assesses whether emotion regulation capacity, inflammation and neuroimmune responses to stress induced in the laboratory prior to the pandemic predict responses to COVID-19-related social isolation and stress in adolescents with CA. We aim to elucidate the mechanisms that enable vulnerable adolescents to maintain or regain good mental health when confronted with COVID-19.

**Methods and analysis** We recruited 79 adolescents aged 16–26 with CA experiences from the Resilience After Individual Stress Exposure study in which we assessed emotion regulation, neural and immune stress responses to an acute stress task. Our sample completed questionnaires at the start of the UK lockdown ('baseline'; April 2020) and three (July 2020) and 6 months later (October 2020) providing crucial longitudinal information across phases of the pandemic progression and government response. The questionnaires assess (1) mental health, (2) number and severity of life events, (3) physical health, (4) stress perception and (5) loneliness and friendship support. We will use multilevel modelling to examine whether individual differences at baseline are associated with responses to COVID-19-related social isolation and stress.

**Ethics and dissemination** This study has been approved by the Cambridge Psychology Research Ethics Committee (PRE.2020.037). Results of the REACT study will be disseminated in publications in scientific peer-reviewed journals, presentations at scientific conferences and meetings, publications and presentations for the general public, and through social media.

## INTRODUCTION

The recent outbreak of a severe acute respiratory syndrome caused by COVID-19 originating in China has escalated globally.[1]

### Strengths and limitations of this study

► Using an integrated biopsychosocial design, we will be able to examine the relations and inter-relations of emotion regulation, neural and immune responses in predicting health and well-being across the trajectory of the COVID-19 lockdown.

► Due to the prospective design, we will be able to examine intraindividual changes in well-being and stress perceptions across the COVID-19 pandemic.

► Due to deep neuroimmune phenotyping prior to COVID-19, we will be able to examine whether individual differences in neuroimmune factors predict well-being and stress perception in response to COVID-19.

► This study relies on retrospective self-report of childhood adversity which may be subject to biased recall.

► Due to the ongoing nature of the pandemic and fluctuations in lockdown restrictions, it may not be possible for us to gauge full recovery from the stress of the pandemic using our current timeline of assessments. For this reason, a fourth wave will be added at a later stage, dependent on funding availability.

Governments have imposed nationwide measures of disease containment and urged their citizens to practice social distancing and isolation. As a consequence, millions of people worldwide are currently experiencing unprecedented periods of social isolation and COVID-related stress[2], likely to result in increased rates of depression, acute and post-traumatic stress and clinical anxiety, especially in already vulnerable populations[3 4] and in demographically mediated ways.[5]

Negative impact of the current pandemic is thought to disproportionately affect populations already at risk for psychopathology, such as youth with a history of childhood adversity (CA).[3 6] CA, comprising emotional, sexual and/or physical abuse, emotional and/or physical neglect, marital distress/conflict, parental mental health problems and/or

parental alcohol dependence, violence and/or aggressive behaviour prior to the age of 16, is one of the strongest predictors of poor mental health and well-being in adulthood.[7-9]

Despite evidence of the detrimental effects of CA on mental health, a significant proportion of individuals show little or no long-term negative sequelae; they display 'resilience'.[10] Currently, our understanding of resilience has been mostly limited to studying outcome after individual-level trauma or commonly experienced catastrophes such as terror attacks or natural disasters. So far, studies examining resilience have either focused on (1) a trait like capacity that precedes adversity, (2) a dynamic process that unfolds during and after adversity or (3) an outcome following adversity. Our recent resilience framework[10] combines these viewpoints and describes that resilience is the dynamic process of positive adaptation to stress, which is aided by resilience factors (traits and states; eg, genetic factors, hormonal levels, brain anatomy, social support), and can be measured in the aftermath of stress in the form of resilient functioning. Resilience in the context of CA, where the stressor has already taken place in childhood, can be assessed through the examination of resilient functioning. This refers to functioning across a range of relevant domains (thoughts, feelings, mood, behaviour, academic ability) that is better than others with similar experiences. A detailed account of this quantification of resilient functioning, and the benefits and drawbacks of this method can be found in Ioannidis *et al.*[11]

The psychological impact of the current pandemic on the general public and modifiable resilience factors that may explain the individual differences in young people's responses have been identified. In China, a cross-sectional study of young people aged 14–35 found that 40.4% of the sample reported having psychological problems and 14.4% presented post-traumatic stress disorder symptoms 2 weeks after the outbreak of COVID-19.[12] A recent study by Li *et al*[13] on 4607 Chinese citizens aged 16 years and over suggested that individuals' cognitive appraisals and perceived severity of the COVID-19 threat was predictive of negative emotional and behavioural responses. Similarly, data from Europe has revealed a positive association between resilience to stress exposure during the COVID-19 lockdown and positive appraisal style in a sample of N=5000 adults.[14] In a large UK sample, personality traits of compassion, conscientiousness, perfectionism and extravertedness were associated with experiencing a more positive global (psychosocioeconomic) impact of the COVID-19 pandemic.[5] In sum, emotion regulation is thought to be a critical resilience factor.[15]

It has been reported that adversity experienced in childhood has potential detrimental effects on emotion regulation capacity and stress regulation through internal biological processes, such as altered inflammatory processes (i.e., changes in circulating lymphocyte count and proinflammatory markers) and neurobiological mechanisms (i.e., increased central executive response

and lower salience network activity).[16-20] During exposure to stress, the body releases proinflammatory markers including high sensitivity C reactive protein (hs-CRP) and cytokines, including interleukin (IL)-1β, IL-6 and tumour necrosis factor (TNF)-α.[21] These cytokines play a key role in acute stress reactivity by activating the hypothalamic–pituitary–adrenal (HPA) axis to release glucocorticoids such as cortisol.[22] In turn, cortisol suppresses the release of cytokines, playing an important anti-inflammatory role in acute stress recovery. While the priming on the immune system following acute stress likely represents an adaptive response, research suggests that acute stress causes an upregulation of inflammatory mediators in circulation and in the amygdala, and a decrease in inflammatory mediators in the medial prefrontal cortex (mPFC)[23]—regions involved in executive functioning and emotion regulation.[24 25] The elevated release of proinflammatory markers in response to a chronic stressor, such as CA, has been associated with both structural and functional alterations in these brain regions.[26] Furthermore, reduced volumes of the amygdala and the mPFC, as well as increased/decreased activation of these regions during executive and affective control tasks may in turn confer vulnerability to mood disorders such as depression and/or anxiety.[27-29]

Research is required to understand how interrelated resilience factors, ranging from 'bottom-up' polygenetic influences to 'top-down' social environmental factors, can facilitate resilient functioning to new stress, such as COVID-19, after CA. The HPA axis, controlling stress reactions and immune functioning, is one system that inextricably intertwines with brain structure and function and may be linked to vulnerable or resilient responses to COVID-19. In the context of CA, cortisol responses to awakening and acute stress response differ for patients with and without psychopathology.[30] Results of a recent meta-analysis revealed that in a healthy population, CA was associated with increased cortisol awakening response and lower baseline cortisol response.[31] In the ongoing Resilience After Individual Stress Exposure (RAISE) study, we investigated how the HPA axis and immune system interrelate with brain structure and function, and social environment to facilitate resilient functioning by measuring emotion regulation capacity and stress responses to a well-validated functional MRI task, the Montreal imaging stress task,[32] in individuals with CA. In the current study, we have the unique opportunity to leverage the RAISE data to investigate how individual differences in emotion regulation and stress responses to previous laboratory-based stressors predicts individual differences in response to COVID-19 stress and isolation. By assessing individuals' responses before, during and after a universally experienced stressor, the Resilience After the COVID-19 Threat (REACT) study will enable us to integrate and examine social, psychological and biological resilience factors across the trajectory and to predict resilience during stress and/or resilient functioning outcomes.

## OBJECTIVES AND HYPOTHESIS

The aim of this study is to examine whether and how social support, emotion regulation capacity, baseline inflammation, and/or neuroimmune responses to laboratory-induced acute stress in adolescents predicts: (1) perceived stress, (2) emotional processing, (3) social behaviour, (4) general mental health and (5) physical health across the trajectory of the COVID-19 restrictions and restructuring.

We hypothesise that social support, resilient emotion regulation and/or stress responses to previous laboratory-based stressors (both in the brain (ie, increased central executive response and lower salience network activity) and in the periphery (ie, lower levels of cortisol, IL-1β, IL-6, TNF-α and hs-CRP)) will be associated with greater resilient functioning as well as better mental and physical health and well-being (lower perceived stress, lower loneliness and increased perception of peer support) across the trajectory of the COVID-19 pandemic.[11 19 29 33–36]

## METHODS AND ANALYSIS

### Study design

All participants have provided written informed consent to complete a set of eleven questionnaires at three time points: at baseline (April 2020), 3 months (July 2020) and 6 months (October 2020) thereafter. By distributing baseline questionnaires in April, a few weeks after the initial lockdown restrictions in the UK were imposed, the social restrictions serve as a timeline for us to measure the wider impact of the pandemic on society. Questionnaires sent at 3 and 6 months thereafter will measure responses as these restrictions are gradually eased. Beyond government measures, the COVID-19 pandemic has a pervasive impact on people's lives due to a substantial and reactive restructuring of society on multiple levels of ecology. The follow-up questionnaires will provide crucial longitudinal measurements, in the context of the impending socio-ecological change, following the phases of the pandemic progression and government response.

### Patient and public involvement

Patient and public involvement was conducted prior to the RAISE study and feedback from the panel was used to improve the quality, relevance and impact of the REACT study. A group of three adolescents participated in the lived experience advisory group to assess our project and the materials we used for recruitment, such as the participant information sheet and consent forms. As a result of this feedback, we made the following changes: (1) we have included a risk protocol in the case that a participant becomes distressed during the completion of the questionnaires (2) within the questionnaires, we have added an evaluation of self-harm and suicidality (3) we have increased the payment for the completion of the study to account for the time burden of the questionnaires, and (4) we have modified the participant information sheet in accordance with their suggestions.

### Participants and recruitment

Participants were recruited from the ongoing RAISE study carried out in the Department of Psychiatry at the University of Cambridge, which involves the completion of an online assessment to assess psychological functioning, emotion regulation capacity and early life experiences as well as an in-unit assessment at Addenbrooke's Hospital to assess neural and immune responses to laboratory-induced acute stress. We contacted 92 individuals from the RAISE Study, of which 79 participants expressed an interest in taking part in REACT and received the first set of questionnaires in April 2020.

Inclusion criteria for the RAISE study were: aged 16–26 years inclusive, able and willing to give informed consent, able to speak, write and understand English, body mass index (BMI) between 18.5 and 29.9 kg/m$^2$, experienced CA (emotional, sexual and/or physical abuse, emotional and/or physical neglect, marital distress/conflict, parental mental health problems and/or parental alcohol dependence, violence and/or aggressive behaviour before the age of 16). Exclusion criteria for the RAISE study were: alcohol or substance use disorder within the past 6 months, current disorders likely to compromise the interpretation of the data (eg, immunological disorders, cardiovascular disorders, endocrine and autoimmune disorders, malignancies or infections or any other condition to be determined by the principal investigator or delegate), current medication likely to compromise the interpretation of immunological data (including, but not limited to, corticosteroids or any other substance to be determined by the principal investigator or delegate).

Participants for the RAISE study were recruited in Cambridge, UK, through advertisements in the general community (e.g., posters in hospitals, colleges, coffee shops) and on social media platforms (e.g., Facebook and Twitter advertisements). We also recruited from an existing database of participants who had previously taken part in the NeuroScience in Psychiatry Network (N=2389) and had consented to being contacted for future studies.

### Study procedure

All RAISE study participants who consented to being recontacted for future studies were approached via email and provided with the participant information sheet. All interested participants were asked to complete online assessments at three time points and were encouraged to contact the study team with any questions that they had.

Participants who expressed interest in taking part received an email with the secure links and instructions to complete the first online consent form and set of questionnaires. The links to the baseline questionnaires were sent at the beginning of April. The same questionnaires and computer tasks were sent 3 and 6 months thereafter to follow-up any changes experienced by the participant.

All three assessments (baseline (April 2020), 3 months (July 2020) and 6 months follow-ups (October 2020)) include the following self-report questionnaires:

Life-Events Questionnaire (LEQ):[37] Number and severity of negative events experienced was assessed with the LEQ. Participants rated 13 major life events which may have occurred during the preceding 18-month period including changes in school, illness, moving house, deaths and friendship difficulties. If affected by the event, respondents were asked to detail what happened and rate it on a scale of 1='very pleasant/happy' to 5='very unpleasant/sad/painful'. A higher score indicates a greater number of adverse life events experienced in the 18 months prior to the assessment.

Physical Health Questionnaire:[38] Physical health was assessed using a 14-item questionnaire that covers four areas of general physical health and areas that are often affected by stress (both acute and chronic), such as gastrointestinal problems, headaches, sleep disturbances and respiratory illness. Higher scores indicates greater somatic health.

The Mood and Feelings Questionnaire (MFQ):[39] The MFQ is a 33-item instrument that was developed to assess mental health and well-being from 2 weeks prior and up to the date of the assessment. The MFQ has shown prognostic validity in both clinical and non-clinical samples.[40 41] Higher sum scores indicate higher levels of depressive symptoms.

The Revised Children's Manifest Anxiety Scale (RCMAS):[42] The RCMAS is a 37-item instrument that assesses physiological anxiety, worry/oversensitivity, social concerns/concentration and total anxiety score. A high score indicates higher levels of anxiety.

Warwick-Edinburgh Mental Well-being Scale (WEMWBS).[43] The WEMWBS assesses responses to 14 positively worded statements to quantify mental well-being in the general population. Each item is graded by the respondent from 'none of the time' to 'all of the time'. Higher scores indicate greater well-being.

Strengths and Difficulties Questionnaire (SDQ) YR1:[44] The SDQ is a 25-item self-report measure, which is divided into five subscales. Four of the subscales measure difficulties and one subscale measures strengths, a prosocial behaviour. The difficulties subscales assess emotional symptoms (internalising symptoms), conduct problems, hyperactivity/inattention and peer relationship problems. The measure has been shown to have good psychometric properties in the age group that will be recruited for the current study (Cronbach's α of 0.80), as well as good sensitivity, specificity and prospective utility. The total difficulties score is calculated by summing the scores of the four difficulties subscales. A high score indicates higher risk of clinically significant problems and can be used to identify likely cases of mental health problems. Conversely, a low score on the prosocial behaviour subscale indicates higher risk of clinically significant problems.

Cambridge Friendship Questionnaire (CFQ): The CFQ is an 8-item questionnaire that assesses the quality and number of friendships. The CFQ has good measurement invariance and external validity, and has demonstrated ecological validity across two samples.[45 46] Higher scores indicate better perceived overall quality of friendships.

Social isolation will be assessed using the Perceived Stress Scale (PSS) and the Revised University of California, Los Angeles Loneliness Scale (R-UCLA):

PSS:[47] The PSS is the most widely used psychological instrument for measuring the perception of stress. It is a 10-item measure used to assess the degree to which situations in one's life are appraised as stressful. Items are designed to assess how unpredictable, uncontrollable, and overloaded respondents find their lives. The scale also includes several direct queries about current levels of experienced stress. A higher sum score indicates higher perceived stress.

R-UCLA:[48] The R-UCLA is a 20-item scale designed to measure subjective feelings of loneliness and social isolation. Participants rate each item on a scale from 1 (never) to 4 (often). Higher scores indicate a greater degree of loneliness.

COVID-19-related stress will be recorded using the COVID-19 Adolescent Symptom & Psychological Experience Questionnaire (CASPE):

CASPE:[49] Emotional, cognitive and social experiences related to COVID-19 will be addressed using the CASPE. The CASPE forms part of the research tracker and facilitator for Assessment of COVID-19 Experiences for Adolescents developed by scientists at the University of Oregon (https://doi.org/10.17605/OSF.IO/PY7VG). The CASPE has a total of 38 items distributed in four major categories: (1) experience related to COVID-19 and symptoms, (2) emotional experience, (3) cognitive experience and (4) social experience.

### Phenotyping

Up to 30 mL of blood was obtained from each participant during the in-unit assessment for the RAISE Study. Measurements of cortisol, blood cytokines and immune cells in response to acute stress will be analysed using partial least squares to determine whether or not resilient adolescents can be distinguished based on immune patterns. By deep neuroimmunephenotyping, we will be able to examine whether individual differences in neuroimmune factors predict well-being and stress perception in response to COVID-19.

### Statistical analysis plan

Responses during and after COVID-19 will be assessed using growth modelling analyses, a statistical technique implemented within a structural equation modelling framework, which allows for the estimation of subject-specific trajectories of change across time for a given set of variables of interest.[50 51] Specifically, implementing growth modelling analyses, we will assess whether individual differences in emotion regulation capacity and/or neural and immune responses to stress in the laboratory at baseline are predictive of individual health and well-being at baseline (intercepts) or trajectories of change before, during and after COVID-19 (slopes). Particularly,

we will estimate intercepts and slopes for mental and physical health and well-being, perceived stress, loneliness and friendship support. In these models, we will add differences in social distancing behaviours and negative life events experienced as covariates, in addition to age, sex and BMI. We will then input these parameters in regression analyses in order to determine whether they are associated with individual differences in emotion regulation capacity and/or neural and immune responses to stress in the laboratory. The analysis of emotion regulation capacity, and/or neural and immune responses to stress in the laboratory at baseline in the RAISE study have been explained in detail in the RAISE study protocol.[52] In short, principle component analyses will be used to derive a factor score for endocrine and inflammatory markers at the different time points (before and after the exposition to a stress task) and imaging data will be analysed using well-established methods for quantification of structural and functional parameters.

## ETHICS AND RISKS

The study has been approved by the Cambridge Psychology Research Ethics Committee (PRE.2020.037). Ethical guidelines will be adhered to throughout the study, including any necessary amendment requests. No restrictions have been placed on the publication of the data and results will be shared with study participants.

All individuals have received the participant information sheet and provided informed consent prior to participation. Some individuals may find the questionnaires (e.g., MFQ) distressing. We ensured that all participants were aware that their participation was voluntary and should they wish to withdraw at any time they would receive a list of local mental health resources and still receive compensation for their time. Participants received £30 compensation for their time for each phase, or £3 for each questionnaire if they chose to withdraw before completion of the study.

Self-injury and suicidality was assessed in the MFQ and disclosure of such was automatically flagged to the research team. The psychiatrists affiliated with the study reviewed participant responses to ascertain imminent risk of self-injury or other risks to the participant. If such risk was identified, then a follow-up conversationtook place to further ascertain the participant's well-being. If the participant appeared at high risk, they were advised to call the first response service (111 option 2) or attend their local Accident and Emergency department. If the risk was moderate, they were advised to contact their general practitioner for support.

All data was anonymised and collected in Research Electronic Data Capture (REDCap) and linked with a unique ID number to personally identifiable information stored in a secure, password encrypted database. Only REACT study team members have access to the database and follow procedures in accordance with the UK Data Protection Act 2018.

## DISSEMINATION

The results of the REACT study will be disseminated through (1) publications in scientific peer reviewed journals, (2) presentations on relevant scientific conferences and meetings, (3) publications and presentations for the general public and (4) through social media.

1. Publications in scientific peer-reviewed journals: Given the importance of this research subject we expect to disseminate the results through publications in high impact journals such as Lancet Psychiatry, JAMA Psychiatry, Molecular Psychiatry, Biological Psychiatry and E-life.
2. Presentations on relevant scientific conferences and meetings: The team will present the results of the study in the form of symposia at conferences and scientific meetings across the world. Anticipated conferences of interest are the Society of Biological Psychiatry conference, Flux, Society for Neuroscience and the Organisation for Human Brain Mapping meeting.
3. Publications and presentations for the general public: The team is passionate about public engagement. Our team have written many publications for the general public and organised the conference 'No nurture, No Chance; Resilience after trauma', which was sold out to 150 members from the general public. In addition, both Cambridge (and nearby London) offer many opportunities for scientific outreach such as the Cambridge Science Festival, Pint of Science (Cambridge) and the London Science Festival.
4. Social media presence: The team has excellent experience with media engagement. Our team is active on social media (twitter), and frequently uses twitter to promote research. Furthermore, the team has excellent lines with established media to promote research. For instance, our team have given many interviews in the traditional media as well as with the 'naked scientist' podcast, and our work has been featured in the Guardian, in blogs and in podcasts.

## CURRENT STATUS

As of October 2020, 79 participants from the RAISE study expressed an interest in taking part in REACT, and completed the baseline questionnaires. From this sample, 77 participants completed the second phase of questionnaires. We anticipate that extra follow-up assessments will be necessary due to the ongoing nature of the pandemic and fluctuations in lockdown restrictions.

## DISCUSSION

As a result of the public health emergency from COVID-19, individuals around the world are experiencing prolonged periods of social isolation and stress. This study will examine how baseline neuro-immune responses and emotion regulation capacity to acute stress in adolescents with CA impact mental health outcomes, emotional processing, social behaviour, physical health

and perceived stress during and after this universal stressor, with the unique approach of leveraging data collected prior to its onset. We hypothesise that during acute stress in the laboratory, increased emotion regulation through increased central executive response and lower salience activity, and reduced baseline cortisol and cytokine levels, either through blunted responsivity or improved down regulation following stress, will be associated with greater mental and physical resilience and lower perceived stress in response to, and during, the societal changes associated with COVID-19. Using an integrative framework, we will examine social, psychological and biological resilience factors across the trajectory of the COVID-19 lockdown. This study aims to characterise the neurobiological mechanisms that contribute to adolescent resilient functioning. We hope that our study will provide important insights for intervention studies, for instance the development of strategies encompassing psychological, cognitive behavioural, somatic or psychopharmacological therapies. Furthermore, our findings may inform the development of current or novel interventions to increase resilience by preventing the development of mental health disorders after adversity.

**Contributors** AJS assisted with study design, data collection and drafted and revised the protocol manuscript. LM-L conceptualised and co-designed the study, oversaw data collection and revised the manuscript. ED, MD, SO and ES assisted with study design and were instrumental to the setup and/or data collection for the study. KI and MK assisted with study design. A-LvH conceptualised and codesigned the study, obtained financial support, oversaw all study procedures and reviewed and revised the manuscript. All authors approved the final manuscript and agree to be accountable for all aspects of the work presented.

**Funding** The REACT study is funded by two grants from the Royal Society to A-LvH (RGF\EA\180029 and RGF\R1\180064). This work was further supported by a Royal Society Dorothy Hodgkin fellowship for A-LvH (DH150176), a Wolfe Health fellowship for LM-L, and a Gates Cambridge Scholarship (OPP1144) for ES.

**Competing interests** None declared.

**Patient and public involvement** Patients and/or the public were involved in the design, or conduct, or reporting, or dissemination plans of this research. Refer to the Methods section for further details.

**Patient consent for publication** Not required.

**Provenance and peer review** Not commissioned; externally peer reviewed.

**ORCID iDs**
Alicia Joanne Smith http://orcid.org/0000-0003-2808-3306
Laura Moreno-López http://orcid.org/0000-0003-3687-7237
Maria Dauvermann http://orcid.org/0000-0002-2873-8512

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
