## [Reviewer comments · BMJ Open]

ARTICLE DETAILS

TITLE (PROVISIONAL)	The REACT study protocol: Resilience after the COVID-19 Threat (REACT) in adolescents.
AUTHORS	Smith, Alicia; Moreno-López, Laura; Davidson, Eugenia; Dauvermann, Maria; Orellana, Sofia; Soneson, Emma; Ioannidis, Konstantinos; Kaser, Muzaffer; van Harmelen, Anne-Laura

VERSION 1 – REVIEW

REVIEWER	Lin Lu Peking University Sixth Hospital, China
REVIEW RETURNED	13-Aug-2020

GENERAL COMMENTS	This article is a protocol of the study to examine how baseline neuro-immune responses and emotion regulation capacity to acute stress in adolescents with childhood adversity influence health and wellbeing across the trajectory of the COVID-19 lockdown in the UK. This study intends to identify resilient factors in adolescents in response to social isolation and COVID-19 related stress, which has important values for the development of novel interventions to prevent mental disorders after adolescent adversity. However, there are several concerns that need to be addressed. 1. In the section of Study procedure, the authors only elaborated the questionnaires to evaluate the adolescent health and wellbeing. However, it is unclear how the authors plan to examine neuro-immune phenotyping in response to acute stress. Additionally, the authors stated “We will use structural equation modelling to examine whether individual differences at baseline are associated with changes in behaviour and responses to social isolation and COVID-19 related stress” in the Abstract. Please describe in more detail in the Methods section. How the authors intend to differentiate social isolation and COVID-19 related stress in the present study?2. The sample size needs to be clarified.3. The authors stated “young adult” in the Title, but stated “adolescents” in the Abstract. Please unify this.
---

REVIEWER	Linda Grabbe Nell Hodgson Woodruff School of Nursing Emory University Atlanta, Georgia USA
REVIEW RETURNED	11-Sep-2020

GENERAL COMMENTS	bmjopen-2020-042824 Review
----------------------------

	The Resilience after the COVID-19 Threat (REACT) study: a prospective study of young adult wellbeing in response to the pandemic. Support for publication This study is a unique opportunity to build on what seems to be a solid set of research data on young persons who have experienced childhood adversity. Results from the authors' ongoing RAISE study include biological measures of emotion regulation and stress responses in this population, and the Covid pandemic offers an opportunity to discover how, under common and real life traumatic circumstances, the youth respond over time to the social isolation and stress of the pandemic, as they first encounter and then maintain themselves during this unfolding strain. This study responds to the need for research regarding the dynamic nature of resilience as it changes for individuals. The 2017 Nature Human Behavior article by Kalisch and colleagues calls for prospective longitudinal research on resilience, and the REACT study is just such work. This will help identify risk factors for deleterious psychological outcomes related to the pandemic, add to the evolution of the resilience framework, and may also help to identify preventive programs and treatment interventions for youth who are both vulnerable and resilient to stress in their lives. Criticisms The questionnaires cover a broad array of dimensions of mental health and do present a burden to the respondents. There are more than 200 questions. The authors should consider the adolescent PROMIS measures for future research. They should also explain why they did not use a resilience measure, such as the widely-used Connor-Davidson Resilience Scale. The pandemic is unlikely to be over in October, so the last survey may possibly be delayed, or a fourth survey added in 2021 if the purpose is to gauge recovery from the common stress of shutdown and threat of the virus. It would be helpful for the authors to share some preliminary RAISE findings, even in summary. While findings may provide insight into the mental health trajectories of youth with histories of adversity and contribute to the science of resilience, the research interpretation should expand and explore the best practices in prevention and healing from trauma.
--	--

VERSION 1 – AUTHOR RESPONSE

Reviewer: 1

Reviewer Name: Lin Lu

Institution and Country: Peking University Sixth Hospital, China

Please state any competing interests or state 'None declared': None declared

This article is a protocol of the study to examine how baseline neuro-immune responses and emotion regulation capacity to acute stress in adolescents with childhood adversity influence health and wellbeing across the trajectory of the COVID-19 lockdown in the UK. This study intends to identify resilient factors in adolescents in response to social isolation and COVID-19 related stress, which has important values for the development of novel interventions to prevent mental disorders after adolescent adversity. However, there are several concerns that need to be addressed.

1. In the section of Study procedure, the authors only elaborated the questionnaires to evaluate the adolescent health and wellbeing. However, it is unclear how the authors plan to examine neuro-immune phenotyping in response to acute stress.

Thank you for your comment. The evaluation of the neuro-immune phenotyping in response to stress is part of the RAISE study protocol; Studying resilience after individual stress exposure (currently under review in BMJ Open). In the RAISE study we examined the neuro-immune responses to stress and their relationship with brain structure and function using the Montreal imaging stress task (MIST) (Dedovic et al., 2005) and venepuncture. Specifically, bloods were acquired at 4 time points to address changes in cortisol, cytokines and immunophenotyping. In the REACT study we will address the effects of COVID-19 in those RAISE study participants willing to take part in the REACT study. We apologise that this was not clear in the protocol and have added details about the neuro-immune phenotyping on page 14 of the manuscript under section '2.5. Phenotyping'.

"2.5. Phenotyping

Up to 30 mL of blood were obtained from each participant during the in-unit assessment for the RAISE Study. Measurements of cortisol, blood cytokines and immune cells in response to acute stress will be analysed using Partial Least Squares (PLS) to determine whether or not resilient adolescents can be distinguished based on immune patterns. By deep neuro-immune-phenotyping we will be able to examine whether individual differences in neuro-immune factors predict wellbeing and stress perception in response to COVID-19."

Dedovic K, Renwick R, Mahani NK, Engert V, Lupien SJ, Pruessner JC. The Montreal Imaging Stress Task: using functional imaging to investigate the effects of perceiving and processing psychosocial stress in the human brain. *The Journal of Psychiatry & Neuroscience*. 2005;30(5):319–25

2. Additionally, the authors stated "We will use structural equation modelling to examine whether individual differences at baseline are associated with changes in behaviour and responses to social isolation and COVID-19 related stress" in the Abstract. Please describe in more detail in the Methods section.

We have now expanded our analysis plan on pages 14-15 under section '2.6. Statistical analysis plan' in the methods section. Please see below.

"2.6. Statistical analysis plan

Responses during and after COVID-19 will be assessed using growth modelling analyses, a statistical technique implemented within a Structural Equation Modelling (SEM) framework, which allows for the estimation of subject-specific trajectories of change across time for a given set of variables of interest (51, 52). Specifically, implementing growth modelling analyses, we will assess whether individual differences in emotion regulation capacity, and/or neural and immune responses to stress in the laboratory at baseline are predictive of individual health and wellbeing at baseline (intercepts) or

trajectories of change before, during and after COVID-19 (slopes). Particularly, we will estimate intercepts and slopes for mental and physical health and wellbeing, perceived stress, loneliness and friendship support. In these models, we will add differences in social distancing behaviours and negative life events experienced as covariates, in addition to age, sex and BMI. We will then input these parameter in regression analyses in order to determine whether they are associated with individual differences in emotion regulation capacity and/or neural and immune responses to stress in the laboratory. The analysis of emotion regulation capacity, and/or neural and immune responses to stress in the laboratory at baseline in the RAISE study have been explained in detail in the RAISE study protocol (Moreno-López et al., in review). In short, principle component analyses (PCA) will be utilized to derive a factor score for endocrine and inflammatory markers at the different time points (before and after the exposition to a stress task) and imaging data will be analysed using well-established methods for quantification of structural and functional parameters.”

Moreno-López L, Sallie SN, Ioannidis K, Kaser M, Schueler K, Askelund AD, et al. The RAISE study protocol; a cross-sectional, multi-level, neurobiological study of studying resilience after individual stress exposure. In review.

3. How do the authors intend to differentiate social isolation and COVID-19 related stress in the present study?

Thank you very much for your question and we apologise that this was unclear in the manuscript. In this study, social isolation and COVID-19 related stress are separate constructs that are recorded using different measures. Social isolation will be assessed using the Perceived Stress Scale and the Revised UCLA Loneliness Scale. Alternatively, COVID-19 related stress will be recorded using the COVID-19 Adolescent Symptom & Psychological Experience Questionnaire. We now mention this and provide details of the questionnaires below and on pages 12-13 of the manuscript.

Social isolation will be assessed using the Perceived Stress Scale and the Revised UCLA Loneliness Scale:

“Perceived Stress Scale (47). The Perceived Stress Scale (PSS) is the most widely used psychological instrument for measuring the perception of stress. It is a 10-item measure used to assess the degree to which situations in one’s life are appraised as stressful. Items are designed to assess how unpredictable, uncontrollable, and overloaded respondents find their lives. The scale also includes several direct queries about current levels of experienced stress. A higher sum score indicates higher perceived stress.

Revised UCLA Loneliness Scale (48). The Revised UCLA Loneliness Scale (R-UCLA) is a 20-item scale designed to measure subjective feelings of loneliness and social isolation. Participants rate each item on a scale from 1 (Never) to 4 (Often). Higher scores indicate a greater degree of loneliness.

COVID-19 related stress will be recorded using the COVID-19 Adolescent Symptom & Psychological Experience Questionnaire:

COVID-19 Adolescent Symptom & Psychological Experience Questionnaire (38). Emotional, cognitive and social experiences related to COVID-19 will be addressed using the COVID-19 Adolescent Symptom & Psychological Experience Questionnaire (CASPE). The CASPE forms part of the research tracker and facilitator for Assessment of COVID-19 Experiences (ACE) for Adolescents developed by scientists at the University of Oregon (<https://doi.org/10.17605/OSF.IO/PY7VG>). The CASPE has a total of 38 items distributed in four major categories: 1) experience related to COVID-19 and symptoms, 2) emotional experience, 3) cognitive experience, and 4) social experience.”

4. The sample size needs to be clarified.

The sample sizes have now been added to page 10 and 15 of the manuscript:

“Participants were recruited from the ongoing RAISE study carried out in the Department of Psychiatry at the University of Cambridge, which involves the completion of an online assessment to assess psychological functioning, emotion regulation capacity and early life experiences as well as an in-unit assessment at Addenbrooke’s Hospital to assess neural and immune responses to laboratory-induced acute stress. We contacted 92 individuals from the RAISE Study, of which 79 participants expressed an interest in taking part in REACT and received the first set of questionnaires in April 2020.”

“As of October 2020, 79 participants from the RAISE study expressed an interest in taking part in REACT, and completed the baseline questionnaires. From this sample, 77 participants completed the second phase of questionnaires. We anticipate that all phases of the study will be completed by November 2020.”

5. The authors stated “young adult” in the Title, but stated “adolescents” in the Abstract. Please unify this.

Thank you very much for highlighting this discrepancy in the paper. The title has now been edited and “adolescents” has been used uniformly throughout the manuscript. The title is now as follows:

“The REACT study protocol: Resilience after the COVID-19 Threat (REACT) in adolescents.”

Reviewer: 2

Reviewer Name: Linda Grabbe

Institution and Country: Nell Hodgson Woodruff School of Nursing, Emory University, Atlanta, Georgia, USA

Please state any competing interests or state ‘None declared’: none

This study is a unique opportunity to build on what seems to be a solid set of research data on young persons who have experienced childhood adversity. Results from the authors’ ongoing RAISE study include biological measures of emotion regulation and stress responses in this population, and the Covid pandemic offers an opportunity to discover how, under common and real life traumatic circumstances, the youth respond over time to the social isolation and stress of the pandemic, as they first encounter and then maintain themselves during this unfolding strain. This study responds to the need for research regarding the dynamic nature of resilience as it changes for individuals. The 2017 Nature Human Behavior article by Kalisch and colleagues calls for prospective longitudinal research on resilience, and the REACT study is just such work. This will help identify risk factors for deleterious psychological outcomes related to the pandemic, add to the evolution of the resilience framework, and may also help to identify preventive programs and treatment interventions for youth who are both vulnerable and resilient to stress in their lives.

Criticisms

1. The questionnaires cover a broad array of dimensions of mental health and do present a burden to the respondents. There are more than 200 questions. The authors should consider the adolescent PROMIS measures for future research.

Thank you for your comment and for bringing to our attention the adolescent PROMIS measures. In future, we will consider the use of adolescent PROMIS measures for our studies.

We took the time restraint of the questionnaires into consideration when designing both the RAISE and REACT study and discussed this with volunteers at the PPI meeting. As suggested at the meeting, we increased the payment for the study to account for this.

In addition, all members of the study team trialled the questionnaires before they were circulated to our participants and found that on average, despite there being more than 200 questions, each phase took no longer than 30 minutes to complete. We additionally highlighted to the participants at the beginning of study that they could withdraw at any time and would be paid in full for the questionnaires that they had completed. Despite this, so far we have had almost no attrition; only two participants that dropped out at the second wave of the study.

2. They should also explain why they did not use a resilience measure, such as the widely-used Connor-Davidson Resilience Scale.

We would like to explain that we did not utilise the Connor Davidson resilience scale as we are interested specifically in resilient functioning after CA as defined by our framework (Kalisch et al., 2017). This framework describes resilience as the dynamic process of positive adaptation to stress, which is aided by resilience factors (e.g. genetic factors, hormonal levels, brain anatomy, social support), and can be measured in the aftermath of stress in the form of resilient functioning. Resilience in the context of CA, where the stressor has already taken place in childhood, can be examined through the examination of resilient functioning. Such resilient functioning after CA refers to functioning across a range of relevant domains - thoughts, feelings, mood, behaviour, academic ability that is better than others with similar CA experiences. For this reason, we included specific measurements of thoughts feelings, mood, and behaviour, as well as CA experiences to calculate level of resilient functioning in our sample.

We have now made this quantification of resilience more clear on page 5-6; where we now write that:

“So far, studies examining resilience have either focussed on [1] a trait like capacity that precedes adversity, [2] a dynamic process that unfolds during and after adversity, or [3] an outcome following adversity. Our recent resilience framework (10) combines these viewpoints and describes that resilience is the dynamic process of positive adaptation to stress, which is aided by resilience factors (traits and states; e.g. genetic factors, hormonal levels, brain anatomy, social support), and can be measured in the aftermath of stress in the form of resilient functioning. Resilience in the context of CA, where the stressor has already taken place in childhood, can be examined through the examination of resilient functioning. Such resilient functioning after CA refers to functioning across a range of relevant domains (thoughts, feelings, mood, behaviour, academic ability) that is better than others with similar CA experiences. A detailed account of this quantification of resilient functioning, and the benefits and drawbacks of this method can be found in Ioannidis et al., (2020).”

3. The pandemic is unlikely to be over in October, so the last survey may possibly be delayed, or a fourth survey added in 2021 if the purpose is to gauge recovery from the common stress of shutdown and threat of the virus.

This is a very important point and we recognise this as a limitation of the study which we have since added to the strengths and limitations section of the manuscript (please see below or on page 4 of the manuscript). We have decided to continue with the third follow-up survey in October as we have already confirmed that participants will take part and be paid for this. However, we are keen to add a final follow-up assessment when the pandemic is over. We are currently exploring funding opportunities to do so.

“Due to the ongoing nature of the pandemic and fluctuations in lockdown restrictions, it may not be possible for us to gauge full recovery from the stress of the pandemic using our current timeline of assessments. For this reason, a fourth wave may be added at a later stage, depending on funding availability.”

4. It would be helpful for the authors to share some preliminary RAISE findings, even in summary.

Thank you for your comment. We apologize that we weren't clear enough that this is a protocol paper, and not an empirical paper. We have now revised the title and abstract to make that clearer. In line with the requirements for protocol papers at BMJ Open, and as per the Editors request, we do not mention results in this protocol paper.

5. While findings may provide insight into the mental health trajectories of youth with histories of adversity and contribute to the science of resilience, the research interpretation should expand and explore the best practices in prevention and healing from trauma.

We thank the reviewer for this suggestion. We have now expanded the insights section on page 19 to include that our basic research lays the foundations on which therapeutic prevention and intervention studies may built on:

“This study aims to characterise the neurobiological mechanisms that contribute to adolescent resilient functioning. We hope that our study will provide important insights for intervention studies, for instance the development of strategies encompassing psychological, cognitive behavioural, somatic or psychopharmacological therapies. Furthermore, our findings may inform the development of current or novel interventions to increase resilience by preventing the development of mental health disorders after adversity.”